# Analyzing lumbar vertebral shape and alignment in female patients with degenerative spondylolisthesis: Comparisons with spinal stenosis and risk factor exploration

**Tomohito Yoshihara, Tadatsugu Morimoto\*, Masatsugu Tsukamoto, Yu Toda, Hirohito Hirata, Takaomi Kobayashi, Satoshi Takashima, Masaaki Mawatari**

Department of Orthopaedic Surgery, Faculty of Medicine, Saga University, Saga, Japan

\* morimot3@cc.saga-u.ac.jp

## Abstract

### Purpose

This study aimed to examine the vertebral body shape characteristics and spondylopelvic alignment in L4 degenerative spondylolisthesis (DS) as well as the risk factors for the development of DS.

### Methods

This cross-sectional study compared vertebral morphology and sagittal spinopelvic alignment in female patients with lumbar DS and lumbar spinal stenosis (LSS). The degree of lumbar lordosis (LL), pelvic incidence (PI), cross-sectional area (CSA), and vertebral body height ratio ($h_a/h_p$) of the lumbar spine were compared using full-length spine radiographs and computed tomography in 60 females with DS and in 60 women with LSS.

### Results

No significant differences in age or body mass index were observed between the two groups; however, the DS and LSS groups significantly differed in PI (mean, 58.9±10.8 vs. 47.2±11.6, $P < 0.001$), L4 CSA (mean, 1,166.2 m$^2$ vs. 1,242.0 m$^2$, $P = 0.002$) and $h_a/h_p$ (mean, 1.134 vs. 1.007, $P < 0.001$). The L4 $h_a/h_p$ was significantly higher in the DS group than in the LSS group. Additionally, LL values were negatively correlated with vertebral L5 CSA in the DS group (r = −0.28, $P < 0.05$). The LSS and DS groups demonstrated positive correlations between LL and L2, L3, and L4 $h_a/h_p$ (r = 0.331, 0.267, and 0.317; $P < 0.01$, < 0.05, and < 0.05, respectively) and between LL and L4 and L5 $h_a/h_p$ (r = 0.333, 0.331; $P < 0.01$, respectively). Multivariate regression analyses revealed that PI and $h_a/h_p$ ratio may be independent predictors of DS development.

**Data Availability Statement:** The underlying data have been deposited at the following DOI: 10.6084/m9.figshare.25459972.

**Funding:** The author(s) received no specific funding for this work.

**Competing interests:** The authors have declared that no competing interests exist.

## Conclusion

The DS group had significantly larger LL, PI, and L4 $h_a/h_p$ and smaller L4 CSA than the LSS group. The lumbar vertebral body shape and sagittal spinopelvic alignment in females might be independent predictors of DS development.

## Introduction

The sagittal spinopelvic alignment influences the prognosis and treatment of adult spinal deformities [1–3]. The parameters used to evaluate sagittal spine pelvic alignment below the lumbar level include lumbar lordosis (LL), sacral slope, pelvic tilt, and pelvic incidence (PI).

LL refers to the anterior curvature of the lumbar spine and is a response to an upright posture in humans that develops during childhood, increases throughout adolescence, and is more common in females than in males [1, 2]. LL is caused by variations in lumbar vertebral morphogenesis, including the ratio of the heights of the anterior and posterior walls of the vertebral body (vertebral body height ratio [$h_a/h_p$], vertebral wedging), height of the intervertebral disc, vertebral cross-sectional area (CSA), and orientation of the lumbar facets [1, 2]. In contrast, PI is the angle of sacral tilt with respect to the pelvis; after bone maturity, PI is an individual-specific constant angle because the movement of the sacroiliac joint is negligible [4]. Lumbar degenerative spondylolisthesis (DS) typically occurs at the L4–L5 level in females aged >50 years with a high PI [5, 6]. Large LL and PI were independent predictors of lumbar DS (ventral slippage of the affected vertebra) [6–8].

In contrast, the CSA represents that of the vertebral body, and a smaller CSA in a growing female vertebral body likely allows for a greater range of spinal motion and facilitates the LL required during pregnancy [4]. However, a large LL may increase the risk of spinal disease, disability, disc height [2], vertebral wedging [5], spondylolysis [6], scoliosis [9], and vertebral fractures [1]. Although the associations between a large LL and DS and between a large LL and CSA have been investigated, an association between vertebral body shape and DS has not been proven [6–8].

In the present study, we hypothesized that DS is associated not only with sagittal spine pelvic alignment, such as LL and PI [6, 9, 10], but also with vertebral body morphology, including wedge-shaped vertebrae and a small CSA.

## Materials and methods

The study protocol was approved by our institutional review board (IRB) for clinical investigations and was conducted in accordance with the Declaration of Helsinki and the Health Insurance Portability and Accountability Act. Written informed consent was obtained from all participants and their parents. All study participants were recruited from the Division of Orthopaedic Surgery of our hospital. Our IRB approved this retrospective review of the patient data (approval number:2021-11-R-06). On February 20, 2022, access was obtained from a database containing records of spinal surgeries conducted at the hospital, and patient data were collected.

### Patients

Because sex differences in CSA and spinopelvic alignments, such as LL and PI, have been reported, sex unification is required. The CSA was reported to be smaller in females than in

males and larger in the LL and PI in females [8, 11]. In addition, it most commonly develops in females and the pelvic morphology differs according to sex [8, 11]. Therefore, only female patients were included in this study. Furthermore, female patients with L4 DS were included in this study because DS is most common at the L4–L5 level and has a higher incidence in female patients than in male patients [12].

Sixty consecutive L4-DS female patients who underwent lumbar posterior fusion at our hospital after January 2018 were included in the study, and 60 consecutive female patients with lumbar spinal stenosis (LSS) who underwent posterior decompression simultaneously were included as the control group for the retrospective iconographic study. Three spinal surgeons diagnosed these diseases based on subjective symptoms, neurological findings, and magnetic resonance imaging findings. All patients underwent sagittal spinal radiography in a standing position without support, spinopelvic computed tomography (CT), or magnetic resonance imaging (MRI) before surgery. Patients with back and leg pain, abnormal neurological findings, and imaging stenosis that could explain these findings were considered for surgery. We diagnosed patients when there was stenosis due to degeneration, such as thickening of the yellow ligament on MRI. L4DS was diagnosed when the L4 vertebra slipped more than 3 mm anteriorly from the L5 vertebra on lateral flexion-extension radiographs of the spine [13].

## Iconographic study

On standing sagittal full-length radiographs of the spine, the angles created by intersecting lines drawn from the midpoint of the femoral heads to the midpoint of the superior endplate of the sacrum and a line perpendicular to the superior endplate of the sacrum were used to measure the PI. The LL was measured as the angle between the superior endplates of L1 and S1. Vertebral CSA was measured in the axial plane at the midportion of the pedicle on the spinopelvic CT images (Fig 1). Vertebral height was measured anteriorly ($h_a$) and posteriorly ($h_p$) in the sagittal plane of the midportion heights in the coronal and axial planes on spinopelvic CT images in the range of L1 L5. Vertebral body height ratio was calculated by dividing the anterior height by the posterior height ($h_a/h_p$) [14].

## Statistical analysis

The mean ± standard deviation of quantitative data was calculated. The Student's t-test was used to compare data between the DS and LSS groups. Qualitative data were expressed as frequencies (percentages). The chi-square test was used to compare data between the DS and LSS groups. To assess the association between DS and iconographic characteristics, multiple multivariate logistic regression analysis was performed to calculate the adjusted odds ratio controlled for age (years, continuous) and body mass index (kg/m$^2$, continuous). The DS (0: absent, 1: present) was used as the dependent variable, and LL (0: <30˚, 1: 30–40˚, and 2: >40˚), PI (0: <50˚, 1: 50–60˚, and 2: >60˚), L4 VCSA (0: 1000–1200 m$^2$, 1: <1000 m$^2$, and 2: >1201 m$^2$), and L4 $h_a/h_p$ (0: >1.1, 1: 1.0–1.1, and 2: <1.0) were used as the independent variables. Statistical significance was set at $P < 0.05$. Data were analyzed using the JMP Pro software version 16 (SAS Institute, Cary, NC, USA). A post hoc power analysis with a Student's t-test (setting of α = 0.05 and 2-tailed) was performed based on PI, LL, L4 VCSA, and L4 $h_a/h_p$ using G* Power 3.1.9.6 (Heinrich-Heine-Universität Düsseldorf, Germany). A post hoc power threshold of 80% was used. Inter-observer agreement between the two board certified spine surgeons of JSSR (The Japanese Society for Spine Surgery and Related Research Reserved), and intra-observer agreement by one reader were analyzed in 30 cases. The intra- and inter-observer agreement for the LL, PI, CSA and $h_a/h_p$ were analyzed.

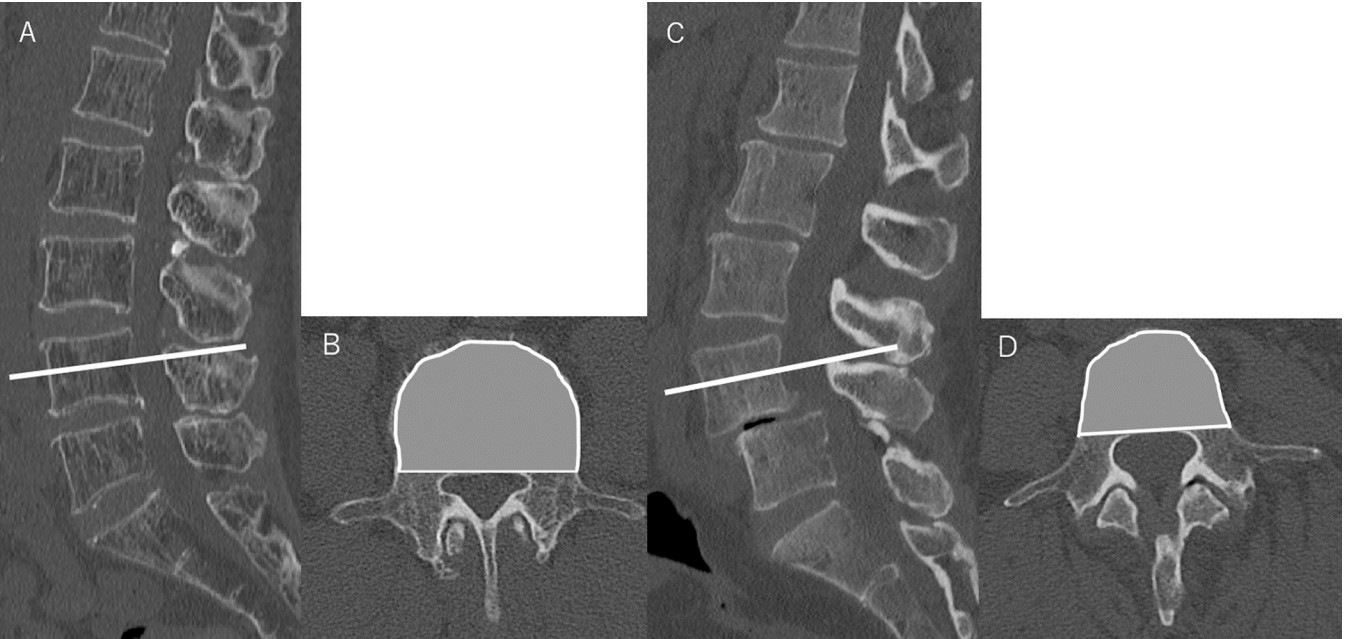

**Fig 1.** CT images of the lumbar spine (A) sagittal and (B) axial for LSS, and (C) sagittal and (D) axial for DS. CT image showing the measurement of vertebral CSA at the lumbar vertebrae. CSA was measured in the sagittal plane in the middle of the L4 pedicle. CT, computed tomography; LSS, lumbar spinal stenosis; DS, degenerative spondylolisthesis; CSA, vertebral cross-sectional area.

## Results

### Reliability of LL and PI, CSA and ha/hp measurements

The kappa values for CSA were 0.91 (95% CI 0.87–0.93) for inter-observer and 0.94 (95% CI 0.92–0.96) for intra-observer. The kappa values for LL were 0.98 (95% CI 0.97–0.99) for inter-observer and 0. 93 (95% CI 0.86–0.97) for intra-observer. The kappa values for PI were 0.95 (95% CI 0.91–0.98) for inter-observer and 0.93 (95% CI 0.85–0.96) for intra-observer. The kappa values for *ha/hp* were 0.97 (95% CI 0.96–0.98) for inter-observer and 0.90 (95% CI 0.86–0.93) for intra-observer. The reliability of the parameter measurements and evaluations was also confirmed.

### Characteristics and parameter comparisons between the two groups

No significant differences in age or body mass index were observed between groups. In contrast, significant differences in PI, L4 CSA, and L4 $h_a/h_p$ were observed between the DS and LSS groups. The L4 $h_a/h_p$ was significantly higher in the DS group than in the LSS group, indicating that the vertebral bodies tended to have a wedge-shaped deformity with an anterior opening. The participant characteristics are presented in Table 1.

### The correlations between each spinal morphology parameter

The LL values negatively correlated with the vertebral L5 CSA in the DS group. The LSS group presented a positive correlation between LL and L2, L3, and L4 $h_a/h_p$, whereas the DS group presemterd a positive correlation between L4 and L5 $h_a/h_p$. However, no significant correlation between the CSA and the $h_a/h_p$ in the DS group was identified. The correlations between each spinal morphological parameter and DS are shown in Table 2.

**Table 1. Characteristics of patients with DS and LSS.**

| Characteristics | DS ($n = 60$) | LSS ($n = 60$) | $P$-value |
|---|---|---|---|
| Age, years | 72.4 ± 5.2 | 73.6 ± 5.4 | 0.447 |
| Body mass index, kg/m$^2$ | 24.8 ± 6.4 | 23.8 ± 5.1 | 0.157 |
| PI, degree | 58.9 ± 10.8 | 47.2 ± 11.6 | <0.001 |
| LL, degree | 40.1 ± 12.0 | 31.1 ± 14.1 | <0.001 |
| L1 CSA, mm$^2$ | 874.9 ± 104.0 | 884.4 ± 93.7 | 0.602 |
| L2 CSA, mm$^2$ | 975.1 ± 112.3 | 996.7 ± 113.3 | 0.296 |
| L3 CSA, mm$^2$ | 1086.7 ± 120.2 | 1130.8 ± 113.8 | 0.041 |
| L4 CSA, mm$^2$ | 1166.2 ± 116.2 | 1242.0 ± 141.4 | 0.002 |
| L5 CSA, mm$^2$ | 1321.8 ± 129.3 | 1353.2 ± 152.1 | 0.225 |
| L1 $h_a/h_p$ | 0.906 ± 0.060 | 0.894 ± 0.051 | 0.227 |
| L2 $h_a/h_p$ | 0.966 ± 0.058 | 0.921 ± 0.065 | <0.001 |
| L3 $h_a/h_p$ | 1.031 ± 0.072 | 0.956 ± 0.057 | <0.001 |
| L4 $h_a/h_p$ | 1.134 ± 0.091 | 1.007 ± 0.087 | <0.001 |
| L5 $h_a/h_p$ | 1.170 ± 0.089 | 1.123 ± 0.102 | 0.007 |

DS, degenerative spondylolisthesis; LSS, lumbar spinal stenosis; PI, pelvic incidence; LL, lumbar lordosis; VCSA, vertebral cross-sectional area; $h_a/h_p$, vertebral body height ratio calculated by dividing the anterior height by the posterior height.

## Multivariate regression analysis of factors contributing to DS development

Multivariate regression analysis with age and body mass index correction revealed that PI and the $h_a/h_p$ ratio were significantly associated with DS diagnosis (Table 3). LL and L4 CSA were not significant independent predictors of DS development.

Post hoc powers of 99.9% at PI, 96.2% at LL, 88.9% at L4 VCSA, and 100% at L4 $h_a/h_p$ were provided. The post hoc power of the parameters was considered sufficient.

**Table 2. Correlations between each parameter and DS scores.**

| | Age | Height | Weight | LL | PI | L1CSA | L2CSA | L3CSA | L4CSA | L5CSA | L1ha/hp | L2ha/hp | L3ha/hp | L4ha/hp |
|---|---|---|---|---|---|---|---|---|---|---|---|---|---|---|
| Height | -0.432 | | | | | | | | | | | | | |
| Weight | -0.421 | 0.349 | | | | | | | | | | | | |
| LL | -0.077 | -0.093 | -0.038 | | | | | | | | | | | |
| PI | 0.284 | -0.291 | -0.202 | 0.413 | | | | | | | | | | |
| L1CSA | 0.380 | 0.095 | 0.052 | 0.083 | 0.021 | | | | | | | | | |
| L2CSA | 0.379 | 0.138 | 0.032 | 0.006 | 0.062 | 0.866 | | | | | | | | |
| L3CSA | 0.423 | 0.096 | 0.085 | -0.059 | 0.125 | 0.842 | 0.864 | | | | | | | |
| L4CSA | 0.408 | 0.033 | 0.047 | -0.146 | 0.085 | 0.754 | 0.758 | 0.851 | | | | | | |
| L5CSA | 0.267 | 0.041 | 0.078 | -0.284 | -0.067 | 0.485 | 0.516 | 0.637 | 0.710 | | | | | |
| L1ha/hp | 0.109 | -0.048 | -0.170 | 0.200 | 0.273 | -0.252 | -0.277 | -0.200 | -0.102 | -0.242 | | | | |
| L2ha/hp | 0.253 | -0.117 | -0.144 | -0.001 | 0.075 | 0.004 | -0.096 | -0.060 | -0.125 | -0.037 | 0.334 | | | |
| L3ha/hp | 0.044 | -0.053 | -0.246 | 0.145 | 0.045 | 0.133 | 0.129 | -0.007 | 0.001 | 0.007 | 0.151 | 0.227 | | |
| L4ha/hp | 0.005 | -0.236 | -0.057 | 0.333 | 0.575 | -0.042 | -0.034 | -0.017 | -0.048 | -0.156 | 0.043 | -0.142 | 0.186 | |
| L5ha/hp | 0.343 | -0.141 | -0.177 | 0.331 | 0.298 | 0.341 | 0.268 | 0.296 | 0.222 | -0.009 | -0.099 | 0.145 | 0.032 | 0.281 |

DS, degenerative spondylolisthesis; LSS, lumbar spinal stenosis; PI, pelvic incidence; LL, lumbar lordosis; CSA, vertebral cross-sectional area; $h_a/h_p$, vertebral body height ratio calculated by dividing the anterior height by the posterior height.

$P < 0.05$ for $|\text{r}| \geq 0.251$. $P < 0.01$ for $|\text{r}| \geq 0.330$.

**Table 3. Results of multivariate logistic regression analyses showing the relationship between DS diagnosis and iconographic parameters.**

| Iconographic parameters | Adjusted OR | 95% CI | P-value |
|---|---|---|---|
| LL | | | |
| <30˚ | 1.00 (Reference) | | |
| 30–40˚ | 1.48 | 0.439–4.982 | 0.528 |
| >40˚ | 1.53 | 0.400–5.879 | 0.532 |
| PI | | | |
| <50˚ | 1.00 (Reference) | | |
| 50–60˚ | 3.40 | 1.055–10.959 | 0.040 |
| >60˚ | 6.68 | 1.385–32.258 | 0.018 |
| L4 CSA | | | |
| <1000 mm$^2$ | 1.00 (Reference) | | |
| 1000–1200 mm$^2$ | 0.45 | 0.047–4.308 | 0.487 |
| >1201 mm$^2$ | 1.00 | 0.105–9.539 | 0.999 |
| L4 $h_a/h_p$ | | | |
| <1.0 | 1.00 (Reference) | | |
| 1.0–1.1 | 2.45 | 0.561–10.725 | 0.233 |
| >1.1 | 14.08 | 2.962–66.928 | <0.001 |

DS, degenerative spondylolisthesis; OR, odds ratio; CI, confidence interval; PI, pelvic incidence; LL, lumbar lordosis; CSA, vertebral cross-sectional area; $h_a/h_p$, vertebral body height ratio calculated by dividing the anterior height by the posterior height.

## Discussion

The main findings of DS in female patients were as follows: (1) the DS group had significantly larger LL, PI, and L4 $h_a/h_p$ (wedge-shaped deformity) and smaller L4 CSA than the LSS group; (2) DS was negatively correlated with LL and L4 CSA and positively correlated with LL and L4 $h_a/h_p$; and (3) multivariate regression analysis indicated that PI and L4 $h_a/h_p$ were significantly more associated with the diagnosis of DS than LL and L4 CSA. To the best of our knowledge, this is the first clinical study to investigate the relationship among $h_a/h_p$, CSA, and DS in female patients.

Generalized joint laxity [11], large LL and PI [1, 2, 11, 15], and vertebral wedge deformity [5] have all been identified as risk factors for the development of DS and are most commonly observed in middle-aged female patients. Surprisingly, the loss of elasticity of the paraspinal ligaments caused by hormonal changes, such as decreased estradiol levels due to menopause or ovariectomy, has also been reported to contribute to the development of degeneration and L4–L5 vertebral slip [16]. In the present study, the DS group demonstrated larger LL, PI, and wedge-shaped deformities with anterior openings than the LSS group, which is consistent with previous reports. In previous reports, the selection of DS cases has varied widely from study to study, with some studies limited to L4–L5 DS cases and others including all DS cases regardless of level; the strength of this study is that it is limited to L4 DS.

As previous evidence suggests that vertebral CSA is a structural determinant of a large LL and spinal flexibility [17], which are risk factors for DS development, we postulated and demonstrated an association between vertebral CSA and DS. Regarding mechanical factors, a small CSA is disadvantageous because it increases stress on the vertebrae during physical activity [17]. This is because for axial compressive loads, the stress in the vertebral body is directly proportional to the applied force and inversely proportional to the CSA [18, 19]. Therefore, a

small CSA may increase stress on the lumbar spine and its segments, including the facet joints, which can lead to DS. Furthermore, as Poorghasamians et al. pointed out, two biomechanical properties associated with the cross-sectional growth of small vertebrae–low strength and high flexibility–simultaneously indicate a possible mechanism for the development and progression of vertebral wedging, which may also lead to DS [17]. Furthermore, the fifth lumbar vertebra is supported between the iliac crests and anchored by the iliolumbar ligament; however, the fourth lumbar vertebra is only partially protected, at which time the sacrum is relatively high, making it susceptible to stress and L4 DS [20].

Longitudinal studies have reported that an increased vertebral wedge associated with a smaller CSA is a predictor of LL progression [17]. In this study, the skeletal structure differed significantly between patients with DS and those with LSS. Patients with DS had a smaller CSA of the lumbar spine. The results of this study also demonstrated three characteristics of DS: larger LL and PI, smaller CSA, and vertebral wedge deformity, similar to those in previous reports. In multivariate analysis, PI and L4 $h_a/h_p$ were identified as independent predictors of DS, but both were positively correlated with LL, suggesting that LL, which was not identified as an independent predictor, may be an important factor in the development of DS. Based on previous reports and the present study, one possible mechanism for the development of DS is that a large PI and small CSA cause an increase in the axial pressure load posteriorly on the vertebral body, resulting in anterior wedging of the vertebral body over time, which in turn causes an increase in LL. In addition, some studies have suggested that immature vertebral plasticity may correct asymmetric growth in response to changes in mechanical stress, as observed in pediatric patients with vertebral fractures caused by leukemia or hypercorticism [21–26]. Therefore, various aspects of DS pathogenesis must be investigated.

The anatomical fact that the L4 CSA is lower in DS patients than in LSS patients alerts us to the following lumbar spine surgery precautions. First, the anterolateral side of the vertebral body may be more likely to be penetrated by the L4 pedicle screw placement (Fig 2). DS may have less grafted bone matrix than LSS during intervertebral fusion. Therefore, DS requires greater attention for intervertebral fusion than LSS.

The present study had several limitations. First, this study was a female-only examination, as sex differences in the CSA, LL, and PI have been reported previously. Therefore, the extent to which the results apply to males and non-Japanese individuals remains unknown. Second,

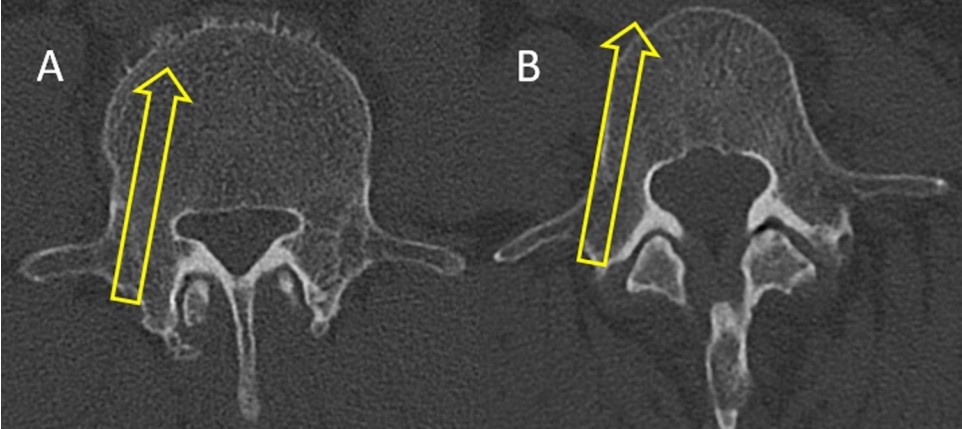

**Fig 2.** CT axial images of the LSS (A) and DS (B). The L4 pedicle screw had a higher risk of deviation from the front lateral side in the DS group than in the LSS group. CT, computed tomography; LSS, lumbar spinal stenosis; DS, degenerative spondylolisthesis.

the LSS group was used as a control group to match propensity scores and cases of the same age, and to clarify variations in surgical precautions, such as pedicle screw placement, due to differences in vertebral body geometry. Another limitation is that this was an observational study, which requires a longitudinal study with a larger number of cases. Further studies are required to corroborate and establish the generalizability of our results to other populations. Further clinical and biomechanical studies are required to confirm these findings.

## Conclusion

Female patients with L4DS had larger LL, PI, and ha/hp, and significantly smaller L4 CSA than those with LSS. Multivariate analysis revealed that the PI and *ha/hp* ratio were independent predictors of DS development. These results suggest that the lumbar vertebral body shape and sagittal spinopelvic alignment in females might be independent predictors of DS development.

## Supporting information

**S1 Checklist. Human participants research checklist.**
(DOCX)

## Acknowledgments

We would like to thank Editage (www.editage.com) for the English language editing.

## Author Contributions

**Conceptualization:** Tomohito Yoshihara, Tadatsugu Morimoto.

**Formal analysis:** Takaomi Kobayashi.

**Investigation:** Tomohito Yoshihara, Takaomi Kobayashi.

**Methodology:** Tomohito Yoshihara, Tadatsugu Morimoto.

**Project administration:** Tadatsugu Morimoto.

**Supervision:** Tadatsugu Morimoto, Masaaki Mawatari.

**Writing – original draft:** Tomohito Yoshihara, Tadatsugu Morimoto, Masatsugu Tsukamoto.

**Writing – review & editing:** Tomohito Yoshihara, Tadatsugu Morimoto, Yu Toda, Hirohito Hirata, Takaomi Kobayashi, Satoshi Takashima, Masaaki Mawatari.

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
