## [Decision Letter · Decision Letter 0]

21 Feb 2024

PONE-D-23-33669Characteristics of lumbar vertebral body shape and sagittal spinopelvic alignment in female patients with lumbar degenerative spondylolisthesisPLOS ONE

Dear Dr. Morimoto,

Thank you for submitting your manuscript to PLOS ONE. After careful consideration, we feel that it has merit but does not fully meet PLOS ONE’s publication criteria as it currently stands. Therefore, we invite you to submit a revised version of the manuscript that addresses the points raised during the review process.

We look forward to receiving your revised manuscript.

Kind regards,

Holakoo Mohsenifar

Academic Editor

PLOS ONE

2. In the online submission form, you indicated that [Raw data were generated at Saga univercity hospital. Derived data supporting the findings of this study are available from the corresponding author TM on request.]. 

3. We note that Figure(s) 1 and 2 in your submission contain copyrighted images. All PLOS content is published under the Creative Commons Attribution License (CC BY 4.0), which means that the manuscript, images, and Supporting Information files will be freely available online, and any third party is permitted to access, download, copy, distribute, and use these materials in any way, even commercially, with proper attribution. For more information, see our copyright guidelines: http://journals.plos.org/plosone/s/licenses-and-copyright.

a. You may seek permission from the original copyright holder of Figure(s) 1 and 2 to publish the content specifically under the CC BY 4.0 license. 

4. Please include your tables as part of your main manuscript and remove the individual files. Please note that supplementary tables (should remain/ be uploaded) as separate ""supporting information"" files.

Reviewers' comments:

Reviewer's Responses to Questions

**Comments to the Author**

1. Is the manuscript technically sound, and do the data support the conclusions?

Reviewer #1: Partly

Reviewer #2: Yes

2. Has the statistical analysis been performed appropriately and rigorously? 

Reviewer #1: No

Reviewer #2: Yes

3. Have the authors made all data underlying the findings in their manuscript fully available?

Reviewer #1: No

Reviewer #2: Yes

4. Is the manuscript presented in an intelligible fashion and written in standard English?

Reviewer #1: No

Reviewer #2: Yes

5. Review Comments to the Author

Reviewer #1: - Title should include the comparison group

- The purpose of the study should be reflected in the title. For example, the risk factors for developing DS are in the purpose but not in the title.

- Abstract conclusion: some of the results on PI variable was reported in the conclusion but not in the results section

- The information provided in the first paragraph is fragmented and lacks cohesion.

- The hypothesis is not clear: part of the hypothesis is that DS is associated with sagittal spine pelvic alignment.

- The information on sample size calculations is very basic, please provide more details about how the sample size was calculated.

- Authors are reporting that “Three spinal surgeons diagnosed these diseases based on subjective

symptoms, neurological findings, and magnetic resonance imaging findings”. These diagnostic criteria should be clarified in order for the study findings to be clinically relevant to health care providers.

- Line 177: there is no such thing as “iliopsoas ligament”. There is iliopsoas muscle and its tendon.

- Lines 212-213: the discussion of the point was very limited. The sentence given is an overstatement of the authors’ findings.

- The author discuss the clinical relevance of their findings.

Reviewer #2: Acceptable report and acceptable flow and well written

Suggest to elaborate on the inter rater reliability of the assessors or how did you ensure that the assessors are reliable in reading the measurements

To mention the validity of the measurement

Suggest to add section about the conclusion of the report

6. PLOS authors have the option to publish the peer review history of their article (what does this mean?). If published, this will include your full peer review and any attached files.

Reviewer #1: No

Reviewer #2: **Yes: **Al Kharusi, Ahmed

---

## [Author Response · Author response to Decision Letter 0]

23 Mar 2024

1. When submitting your revisions, we will address these additional requirements.

Response: Thank you for providing the information on the submission styles.

We have reviewed and modified the style requirements of PLOS ONE.

2. In the online submission form, you indicated that [raw data were generated at the Saga University City Hospital. Derived data supporting the findings of this study are available from the corresponding author, TM, upon request.]. 

All PLOS journals now require that all data underlying the findings described in their manuscript be freely available to other researchers, either 1. In a public repository, a 2. within the manuscript itself, or 3. Available as supplementary information. 

This policy applies to all data except where public deposition breaches compliance with the protocol approved by the research ethics board. If your data cannot be made publicly available for ethical or legal reasons (e.g., public availability would compromise patient privacy), please explain your reasons for resubmission, and your exemption request will be escalated for approval. 

Response: We have attached the research data that can be made publicly available.

3. Figures (s) 1 and 2 in your submission contain copyrighted images. All PLOS content has been published under the Creative Commons Attribution License (CC BY 4.0), which means that the manuscript, images, and Supporting Information files will be freely available online, and any third party is permitted to access, download, copy, distribute, and use these materials in any way, even commercially, with proper attribution. For more information, see our copyright guidelines: http://journals.plos.org/plosone/s/licenses-and-copyright.

a. You may seek permission from the original copyright holder of Figure(s) 1 and 2 to publish the content specifically under the CC BY 4.0 license. 

“I request permission for the open-access journal PLOS ONE to publish XXX under the Creative Commons Attribution License (CCAL) CC BY 4.0 (http://creativecommons.org/licenses/by/4.0/). Please be aware that this license allows unrestricted use and distribution by third parties, even commercially. Please reply and provide explicit written permission to publish XXX under a CC BY license and complete the attached form.”

b. If permission cannot be obtained from the original copyright holder to publish these figures under the CC BY 4.0 license or if the copyright holder’s requirements are incompatible with the CC BY 4.0 license, please either i) remove the figure or ii) supply a replacement figure that complies with the CC BY 4.0 license. Please check the copyright information for all replacement figures and update the figure captions using the source information. If applicable, please specify in the figure caption text when a figure is similar but not identical to the original image and is, therefore, for illustrative purposes only.

Response: Thank you for pointing this out. The figures used in this study were CT images of surgical cases at our hospital, and prior approval for publication was obtained from the patient using an informed consent form. 

4. Please include the tables as part of your main manuscript and remove individual files. Please note that supplementary tables (should remain or be uploaded) are separate Supporting Information files.

Response: Thank you for your instructions. We have deleted the Table file.

 

Response to Reviewer 1

We thank the esteemed reviewer for their helpful comments. 

＞Title should include the comparison group.

＞The purpose of the study should be reflected in the title. For example, the risk factors for developing DS are in the purpose but not in the title.

Response: Thank you for pointing this out. The comparison groups and objectives have been added to the title. 

Analysis of lumbar vertebral shape and alignment in female patients with degenerative spondylolisthesis: Comparison with spinal stenosis and risk factor exploration.

＞Abstract conclusion: some of the results on PI variable was reported in the conclusion but not in the results section.

Response: Thank you for pointing this out.

Multivariate regression analysis with age and body mass index correction revealed that PI and the ha/hp ratio were significantly associated with DS diagnosis. This has been documented in the results section (Lines 160-161).

PI was added to the Results section (Lines 27,139).

＞The information provided in the first paragraph is fragmented and lacks cohesion. 

Response: Thank you for providing us with the opportunity to clarify this point. We have revisited the said paragraph and added the following sentence, which we hope is sufficient:

The parameters used to evaluate sagittal spine pelvic alignment below the lumbar level include lumbar lordosis (LL), sacral slope, pelvic tilt, and pelvic incidence (PI) (Lines 43-44).

＞The hypothesis is not clear: part of the hypothesis is that DS is associated with sagittal spine pelvic alignment.

Response: Thank you for the comment. We have revised the hypothesis as follows:

“In the present study, we hypothesized that DS is associated not only with sagittal spine and pelvic alignment, such as LL and PI [6, 9-10], but also with vertebral body morphology, including wedge-shaped vertebrae and small CSA.” in the Introduction section (Lines 61-63).

＞The information on sample size calculations is very basic.

Response: Thank you for providing us with the opportunity to clarify this point.

This study is novel; thus, we conducted a post-hoc power analysis to further support our findings. We included a detailed explanation of this analysis in the revised manuscript. 

A post hoc power analysis with a Student’s t-test (setting of α = 0.05 and 2-tailed) was performed based on PI, LL, L4 VCSA, and L4 ha/hp using G* Power 3.1.9.6 (Heinrich-Heine-Universität Düsseldorf, Germany). A post hoc power threshold of 80% was used. (Lines 121-123)

Post hoc powers of 99.9% at PI, 96.2% at LL, 88.9% at L4 VCSA, and 100% at L4 ha/hp were provided. The post hoc power of the parameters was considered sufficient. (Lines 169-170)

> Authors are reporting that “Three spinal surgeons diagnosed these diseases based on subjective symptoms, neurological findings, and magnetic resonance imaging findings”. These diagnostic criteria should be clarified in order for the study findings to be clinically relevant to health care providers. 

Response: Thank you for providing us with the opportunity to clarify this point.

The following text was inserted in Lines 87-91:

Patients with back and leg pain, abnormal neurological findings, and imaging stenosis were considered for surgery. We diagnosed the patients with stenosis due to degeneration, such as thickening of the yellow ligament on MRI. L4DS was diagnosed when the L4 vertebra slipped more than 3 mm anteriorly from the L5 vertebra on lateral flexion-extension radiographs of the spine [13].

> Line 177: there is no such thing as “iliopsoas ligament”. There is iliopsoas muscle and its tendon.

Response: Thanks for pointing out the typo.

We misspelled “ilio-lumbar ligament” as “iliopsoas ligament” (Line 198). The “ilio-lumbar ligament” was taken from the text of Fitzgerald et al. (reference 20).

>Lines 212-213: the discussion of the point was very limited. The sentence given is an overstatement of the authors’ findings. 

Response: Thank you for pointing this out.

We have removed the following (Lines 232-233): Furthermore, these results suggest that a large ha/hp ratio and PI may be independent predictors of future lumbar spine deformities and diseases, including DS.

 

Response to Reviewer 2

We thank the reviewers for their helpful comments. We hope that these changes have improved the quality of our work. The individual conclusions are as follows:

>Suggest to elaborate on the inter rater reliability of the assessors or how did you ensure that the assessors are reliable in reading the measurements.

>To mention the validity of the measurement.

Response: Thank you for providing us with the opportunity to clarify this point.

The following text was inserted (Lines 123-126):

Inter-observer agreement between the two board certificated spine surgeons of JSSR (The Japanese Society for Spine Surgery and Related Research Reserved), and intra-observer agreement by one reader were analyzed in 30 cases. The intra- and inter-observer agreement for the LL, PI, CSA and ha/hp were analyzed.

The following text was inserted (Lines 130-135):

The kappa values for CSA were 0.91 (95% CI 0.87–0.93) for inter-observer and 0.94 (95% CI 0.92–0.96) for intra-observer. The kappa values for LL were 0.98 (95% CI 0.97–0.99) for inter-observer and 0. 93 (95% CI 0.86–0.97) for intra-observer. The kappa values for PI were 0.95 (95% CI 0.91–0.98) for inter-observer and 0.93 (95% CI 0.85–0.96) for intra-observer. The kappa values for ha/hp were 0.97 (95% CI 0.96–0.98) for inter-observer and 0.90 (95% CI 0.86–0.93) for intra-observer.

The reliability of parameter measurements and evaluations in this study was confirmed.

>Suggest to add section about the conclusion of the report.

Response: Thank you for your helpful comment.

We added a separate Conclusion section (Lines 235-239).

Conclusion

Female patients with L4DS had larger LL, PI, and ha/hp, and significantly smaller L4 CSA than those with LSS. The multivariate analysis revealed that the PI and ha/hp ratio were independent predictors of DS development. These results suggest that the lumbar vertebral body shape and sagittal spinopelvic alignment in females may be independent predictors of DS development.

---

## [Editor Report · Decision Letter 1]

27 Mar 2024

Analyzing lumbar vertebral shape and alignment in female patients with degenerative spondylolisthesis: Comparisons with spinal stenosis and risk factor exploration

PONE-D-23-33669R1

Dear Dr. Morimoto,

We’re pleased to inform you that your manuscript has been judged scientifically suitable for publication and will be formally accepted for publication once it meets all outstanding technical requirements.

Kind regards,

Holakoo Mohsenifar

Academic Editor

PLOS ONE
---

## [Editor Report · Acceptance letter]

3 Apr 2024

PONE-D-23-33669R1 

PLOS ONE

Dear Dr. Morimoto, 

I'm pleased to inform you that your manuscript has been deemed suitable for publication in PLOS ONE. Congratulations! Your manuscript is now being handed over to our production team.

Kind regards, 

on behalf of

Dr. Holakoo Mohsenifar 

Academic Editor

PLOS ONE